# MoLA: Motion Generation and Editing with Latent Diffusion Enhanced by Adversarial Training

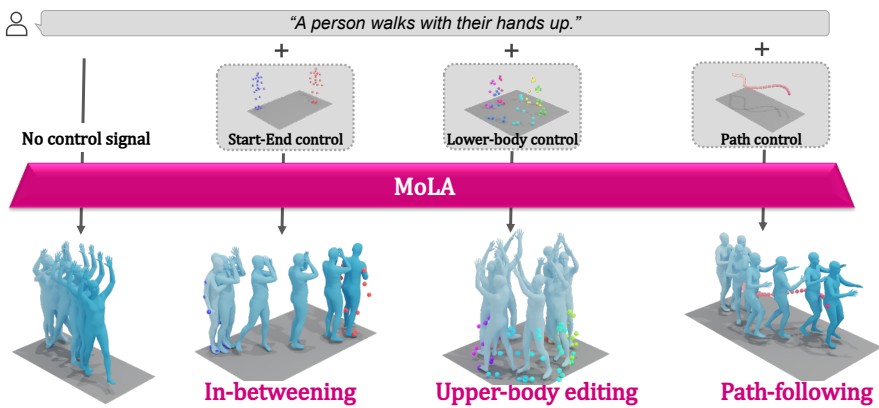

Figure 1. MoLA achieves fast and high-quality human motion generation given textual descriptions while enabling motion editing applications. With MoLA, we can deal with various types of motion editing tasks in a single framework.

## Abstract

In text-to-motion generation, controllability as well as generation quality and speed has become increasingly critical. The controllability challenges include generating a motion of a length that matches the given textual description and editing the generated motions according to control signals, such as the start-end positions and the pelvis trajectory. In this paper, we propose MoLA, which provides fast, high-quality, variable-length motion generation and can also deal with multiple editing tasks in a single framework. Our approach revisits the motion representation used as inputs and outputs in the model, incorporating an activation variable to enable variable-length motion generation. Additionally, we integrate a variational autoencoder and a latent diffusion model, further enhanced through adversarial training, to achieve high-quality and fast generation. Moreover, we apply a training-free guided generation framework to achieve various editing tasks with motion control inputs. We quantitatively show the effectiveness of adversarial learning in text-to-motion generation, and demonstrate the applicability of our editing framework to multiple editing tasks in the motion domain.

## 1. Introduction

Human motion synthesis from text is an emerging task with highly relevant applications in fields such as multimedia production and computer animation. For example, an animator might wish to create or edit a motion prototype to verify their artistic intent before time-consuming animation or motion capture commences, generate smooth in-between motion between two motion capture clips, or generate specific motion that follows a predefined trajectory. In Figure 1, we demonstrate results of our approach on such motion generation and editing tasks. To be useful in those real-world applications, a method has to excel in three domains: (1) motion quality, which encompasses both the general motion quality and the adherence to the textual description; (2) fast inference time; (3) efficient motion editing.

Several recent works have attempted to address these desired properties: Methods based on vector quantization (VQ), such as T2M-GPT [38], MoMask [11], MMM [25], ParCo [43], and BAMM [24], achieve impressive generation quality by compressing human motion into discrete tokens and then sampling those tokens to synthesize motion. Diffusion-based methods in data space, such as MDM [32], provide impressive flexibility with regard to quality and mo-

tion editability. To further improve motion quality and inference time, latent-space based methods such as MLD [2] or MotionMamba [41] operate in a learned continuous latent space.

However, no state-of-the-art method excels in all three domains necessary for real-world applications. For example, while latent-space based approaches such as VQ-based methods or MLD achieve impressive motion quality and fast inference time, they cannot edit a given motion sequence in a training-free manner. In contrast, data-space based methods such as MDM are able to edit motion in a training-free manner, which, however, comes at the cost of slow inference time and lower generation quality. Moreover, while these models excel at motion generation, they require users to manually specify the motion length instead of automatically determining it from the textual input. This often necessitates length estimation or iterative adjustment, which limits their flexibility. For instance, MoMask [11] uses a length estimator during inference to predict motion length based on text input. However, inaccurate predictions may result in significant motion drift [34].

To close this gap, we propose **MoLA**, **Mo**tion Generation and Editing with **L**atent Diffusion Enhanced by **A**dversarial Training. We revisit the representation of motion features used as inputs and outputs of the model and introduce an activation variable that characterizes the length of the motion. We utilize a variational auto-encoder (VAE) [18] for its continuous latent space, which allows training-free editing. This is in contrast to discrete latent spaces that do not allow for training-free motion editing. We also enhance the VAE training with adversarial learning, which has been shown to work well in the image [6, 14, 27] and audio [21] domains. We empirically show that our model achieves variable-length motion generation aligned with textual descriptions and high generation performance on a commonly used dataset [10]. We also demonstrate that training-free guided diffusion can enable multiple motion editing tasks such as path-following, in-betweening, and upper body editing. These experiments also highlight the significant improvement in speed and performance of our model compared to the performance of existing training-free motion editing models (see Figure 2).

The main contributions of this paper are threefold. First, we introduce an activation variable into the motion representation and show that our method can perform variable-length motion generation conditioned on text. Second, we propose a new continuous latent-based motion generation model that introduces adversarial training into the motion VAE and quantitatively show that it significantly pushes the limits of existing continuous-based methods. Finally, we demonstrate that our model not only shows high generation performance but also can deal with various types of motion editing tasks in a training-free manner.

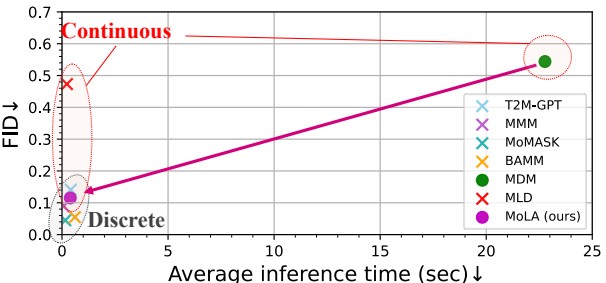

Figure 2. Comparison of inference cost, generation performance, and editability for text-to-motion methods on HumanML3D dataset. ● means a method that can edit motion in a training-free manner, and × means a method that cannot edit motion in a training-free manner. All tests are performed on the same NVIDIA A100 GPU. The pink arrow in the figure indicates that our method significantly extends the performance boundaries (in terms of generation quality and speed) of methods categorized as enabling training-free editing.

## 2. Related Work

### 2.1. Motion Generation

Text-to-motion generation technology has made rapid progress with the diffusion models and VQ-based models. MDM [32] and MotionDiffuse [40] adopt diffusion models for motion generation, which leads to better performance in terms of generation quality. However, these methods directly apply diffusion processes to raw motion sequences, thus resulting in slow generation. MLD [2] mitigates this issue by adopting a diffusion model in a low-dimensional latent space provided by a VAE trained on motion data, inspired by latent diffusion models [27]. VQ-based models have been studied as well, inspired by the success of VQ-based models in image generation [1, 8, 36]. In this approach, a VQ-VAE model is first trained on motion data to acquire discrete motion representations, and deep generative models are then applied to generate sequences of discrete representations (also called tokens). T2M-GPT [38], AttT2M [42], MotionGPT [15] and ParCo [43] utilize autoregressive (AR) models to generate motion tokens. However, AR models are slow in inference because motion tokens are generated sequentially. To address this issue, M2DM [19] and DiverseMotion [22] apply discrete diffusion models to motion tokens in a latent space, whereas MMM [25], MoMask [11] and BAMM [24] adopt a mask prediction model.

### 2.2. Motion Editing

Motion editing has attracted much research interest as well. MDM [32] demonstrated upper body editing and motion in-betweening by applying diffusion inpainting to motion data in both the spatial and temporal domains. LGD [29] demon-

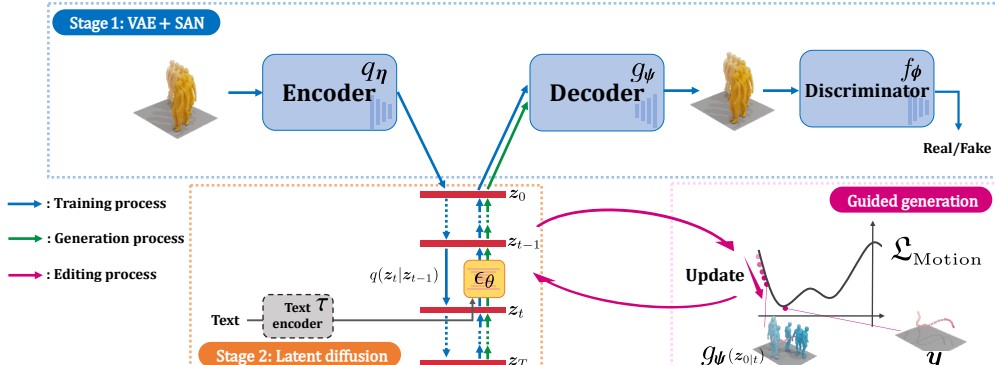

Figure 3. The overall framework of MoLA. Stage 1: A motion VAE enhanced by adversarial training learns a low-dimensional latent representation of diverse motion sequences. Stage 2: A text-conditioned latent diffusion model leverages this representation for fast and high-quality text-to-motion generation. Guided generation: During inference, a gradient-based method minimizes a loss function $\mathcal{L}_{\text{Motion}}$ for each desired editing task, enabling multiple motion editing tasks within a unified framework.

strated path-following motion generation with a guided diffusion that utilizes multiple samples from a suitable distribution to reduce bias. GMD [16] guides the position of the root joint to control motion trajectories. OmniControl [35] controls any joints at any time by guiding a pretrained motion diffusion model with an analytic function. DNO [17] optimizes the diffusion latent noise of a pretrained text-to-motion model with user-provided criteria in the motion space and achieves multiple editing tasks. However, these methods employ data-space diffusion models similar to MDM and have not been demonstrated with a latent diffusion model. Recently, MMM [25] demonstrated motion editing by placing masked tokens in the place that needs editing and applying the mask prediction framework, and MotionLCM [4] proposed a fast controllable motion generation framework by introducing latent consistency distillation and the motion ControlNet [39] manipulation in the latent space.

## 3. Method

The goal of this study is to develop a framework for fast and high-quality text-guided motion generation and to deal with multiple control tasks in a training-free way. To achieve this, we propose the following training and inference tricks (I)-(IV): (I) We review the representation of motion features to achieve improved accuracy and variable length motion generation. We select the necessary features to reduce the burden on our model's VAE encoder and add an activation variable to the representation to determine the motion length (Section 3.1). (II) We train a motion VAE enhanced by adversarial training to achieve high generation performance (Section 3.2). (III) To reduce computational complexity while simultaneously enabling high-quality text-driven motion generation, we train a text-conditioned diffusion model on the low-dimensional latent space obtained

by VAE model (Section 3.3). (IV) Guided generation: Adopting training-free guided generation in inference enables multiple editing functions required in motion generation without additional training (Section 3.4). The outline of our text-to-motion generation model, MoLA, is shown in Figure 3.

### 3.1. Motion representation

We build upon the motion representation used in [10, 26] and improve the representation to enable variable length generation and improve performance in our framework. The pose representation used in [10, 26] is expressed as $\boldsymbol{m} \in \mathbb{R}^{(4+12N_j+4)\times L}$, where $N_j$ is the number of joints. A motion can be represented as a sequence of this representation. The $i$-th ($1 \leq i \leq L$) pose is defined as follows:

$$\boldsymbol{m}^i = [\dot{r}_a^i, \dot{r}_x^i, \dot{r}_z^i, r_y^i, (\boldsymbol{j}_p^i)^\top, (\boldsymbol{j}_r^i)^\top, (\boldsymbol{j}_v^i)^\top, (\boldsymbol{c}^i)^\top]^\top \quad (1)$$

where $\dot{r}_a^i \in \mathbb{R}$ is root angular velocity along the $Y$-axis, $\dot{r}_x^i$ and $\dot{r}_z^i \in \mathbb{R}$ are root linear velocities on $XZ$-plane, $r_y^i$ is root height, $\boldsymbol{j}_p^i \in \mathbb{R}^{3(N_j-1)}$ is local joints positions, $\boldsymbol{j}_r^i \in \mathbb{R}^{6(N_j-1)}$ is rotations in root space, $\boldsymbol{j}_v^i \in \mathbb{R}^{3N_j}$ is velocities and $\boldsymbol{c}^i \in \mathbb{R}^4$ is binary foot-ground contact features by thresholding the heel and toe joint velocities. To achieve variable-length motion generation during stage 2 inference (explained in Section 3.3), we concatenate an activation variable $a^i \in \mathbb{R}$ for the $i$-th pose to Equation (1) as $[(\boldsymbol{m}^i)^\top, a^i]^\top$. Additionally, to reduce the burden on the encoder in Section 3.2, we remove the substantially redundant information $\boldsymbol{j}_v$ and $\boldsymbol{c}$ from Equation (1) as $\tilde{\boldsymbol{m}}^i = [\dot{r}_a^i, \dot{r}_x^i, \dot{r}_z^i, r_y^i, (\boldsymbol{j}_p^i)^\top, (\boldsymbol{c}^i)^\top]^\top$, and then we set the vector for the encoder input in Section 3.2 as follows:

$$\boldsymbol{x}^i = [(\tilde{\boldsymbol{m}}^i)^\top, a^i]^\top = [\dot{r}_a^i, \dot{r}_x^i, \dot{r}_z^i, r_y^i, (\boldsymbol{j}_p^i)^\top, (\boldsymbol{j}_r^i)^\top, a^i]^\top. \quad (2)$$

We employ $\boldsymbol{x} = [\boldsymbol{x}^1, \ldots, \boldsymbol{x}^L] \in \mathbb{R}^{N \times L}$ as motion representation in our model training, where $N = 4 + 9N_j + 1$.[1]

## 3.2. Stage 1: Continuous Motion Latent Representation with Adversarial Training

### 3.2.1. Learning a motion latent representation with VAE-GAN

We propose a motion variational autoencoder (VAE), enhanced by adversarial training, to learn a low-dimensional latent representation for diverse human motion sequences. Assume an observed motion $\boldsymbol{x} \in \mathbb{R}^{N \times L}$, where $N$ and $L$ denote raw motion data dimension per frame and motion length, respectively. To learn such a latent representation, we first define a latent variable $\boldsymbol{z} \in \mathbb{R}^{d_z \times d_l}$, which is assumed to generate data sample $\boldsymbol{x}$, where $d_z, d_l \in \mathbb{N}$. The generative process is modeled as $\boldsymbol{x} \sim p_{\boldsymbol{\psi}}(\boldsymbol{x}|\boldsymbol{z})$ with a prior $p(\boldsymbol{z})$. The prior is assumed to be a standard Gaussian distribution, i.e., $p(\boldsymbol{z}) = \mathcal{N}(\boldsymbol{0}, \boldsymbol{I})$. We model the conditional distribution as a Gaussian distribution: $p_{\boldsymbol{\psi}}(\boldsymbol{x}|\boldsymbol{z}) = \mathcal{N}(g_{\boldsymbol{\psi}}(\boldsymbol{z}), \sigma^2 \boldsymbol{I})$ with $g_{\boldsymbol{\psi}} : \mathbb{R}^{d_z \times d_l} \rightarrow \mathbb{R}^{N \times L}$ and $\sigma^2 \in \mathbb{R}_+$, where $\mathbb{R}_+$ indicates the set of all positive real numbers. As in a usual VAE, we introduce an approximated posterior, which is modeled by $q_{\boldsymbol{\eta}}(\boldsymbol{z}|\boldsymbol{x}) : \mathbb{R}^{N \times L} \rightarrow \mathbb{R}^{d_z \times d_l}$. As a result, the VAE consists of an encoder and a decoder, parameterized by $\boldsymbol{\eta}$ and $\boldsymbol{\psi}$, respectively. The objective function for the VAE is formulated as the negative evidence lower bound (negative ELBO) per sample $\boldsymbol{x}$, which is a weighted summation of mean squared error and KL regularization terms:

$$
\begin{aligned}
\mathcal{J}_{\text{VAE}}(\boldsymbol{\psi}, \boldsymbol{\eta}; \boldsymbol{x}) = \mathbb{E}_{q_{\boldsymbol{\eta}}(\boldsymbol{z}|\boldsymbol{x})}[\|\boldsymbol{x} - g_{\boldsymbol{\psi}}(\boldsymbol{z})\|_2^2] \\
+ \lambda_{\text{reg}} D_{KL}(q_{\boldsymbol{\eta}}(\boldsymbol{z}|\boldsymbol{x}) \| p(\boldsymbol{z})), \quad (3)
\end{aligned}
$$

where $\lambda_{\text{reg}}$ is a hyperparameter for balancing the two terms. The encoder $q_{\boldsymbol{\eta}}$ can be trained to produce a low-dimensional latent representation, and the decoder $g_{\boldsymbol{\psi}}$ can also be trained to accurately reconstruct the input motion sequences from the latent representations. Besides, we separate the decoder output into two components, denoted as $g_{\boldsymbol{\psi}}(\boldsymbol{z}) = [(\boldsymbol{m}'(\boldsymbol{z}))^\top, \boldsymbol{a}'(\boldsymbol{z})]^\top$, where $\boldsymbol{m}'(\boldsymbol{z}) \in \mathbb{R}^{(N-1) \times L}$ and $\boldsymbol{a}'(\boldsymbol{z}) \in \mathbb{R}^L$ correspond to the original motion and proposed activation variable. We adopt a binary cross entropy (BCE) loss for the activation variable. The following $\mathcal{J}_{\text{MotionVAE}}$ is used as the loss function for the VAE part:

$$
\begin{aligned}
\mathcal{J}_{\text{MotionVAE}}(\boldsymbol{\psi}, \boldsymbol{\eta}; \boldsymbol{x}) = \mathbb{E}_{q_{\boldsymbol{\eta}}(\boldsymbol{z}|\boldsymbol{x})}[\|\boldsymbol{m} - \boldsymbol{m}'(\boldsymbol{z})\|_2^2 \\
+ \lambda_{\text{act}} \mathcal{L}_{BCE}(\boldsymbol{a}, \boldsymbol{a}'(\boldsymbol{z}))) ] + \lambda_{\text{reg}} D_{KL}(q_{\boldsymbol{\eta}}(\boldsymbol{z}|\boldsymbol{x}) \| p(\boldsymbol{z})),
\end{aligned}
$$
(4)

where $\lambda_{\text{act}}$ is a hyperparameter for weighting the BCE loss term.

---

[1] In practice, during training, we pad zeros to $\boldsymbol{m}$ or $\tilde{\boldsymbol{m}}$ in inactive frames used to align sequence lengths. Then, $a^i = 0$ is assigned to padded frames, and $a^i = 1$ is assigned to frames containing motion data.

To achieve high-quality generation, we need to push the limits of compression. Hence, we propose incorporating adversarial training into the motion VAE (c.f. [21, 27]). More specifically, we introduce a discriminator, denoted as $f_{\boldsymbol{\phi}} : \mathbb{R}^{N \times L} \rightarrow \mathbb{R}$, that aims to distinguish real and reconstructed motions. The adversarial training is formulated as a two-player optimization between the VAE and the discriminator. The discriminator is trained by the maximization of $\mathbb{E}_{p(\boldsymbol{x})} \mathcal{L}_{\text{GAN}}(\boldsymbol{\phi}; \boldsymbol{\psi}, \boldsymbol{\eta}, \boldsymbol{x})$ with respect to $\boldsymbol{\phi}$, where

$$
\begin{aligned}
\mathcal{L}_{\text{GAN}}(\boldsymbol{\phi}; \boldsymbol{\psi}, \boldsymbol{\eta}, \boldsymbol{x}) = \min\{0, -1 + f_{\boldsymbol{\phi}}(\boldsymbol{x})\} \\
+ \mathbb{E}_{q_{\boldsymbol{\eta}}(\boldsymbol{z}|\boldsymbol{x})} \left[ \min\{0, -1 - f_{\boldsymbol{\phi}}(g_{\boldsymbol{\psi}}(\boldsymbol{z}))\} \right].
\end{aligned}
$$
(5)

We formulate the overall loss for the VAE as the sum of the negative ELBO and adversarial loss,

$$
\min_{\boldsymbol{\phi}, \boldsymbol{\psi}} \mathbb{E}_{p(\boldsymbol{x})} \left[ \mathcal{J}_{\text{MotionVAE}}(\boldsymbol{\psi}, \boldsymbol{\eta}; \boldsymbol{x}) + \lambda_{\text{adv}} \mathcal{J}_{\text{GAN}}(\boldsymbol{\psi}, \boldsymbol{\eta}; \boldsymbol{\phi}, \boldsymbol{x}) \right],
$$
(6)

where $\lambda_{\text{adv}}$ is a positive scalar that adjusts the balance between the two terms, and

$$
\mathcal{J}_{\text{GAN}}(\boldsymbol{\psi}, \boldsymbol{\eta}; \boldsymbol{\phi}, \boldsymbol{x}) = - \mathbb{E}_{q_{\boldsymbol{\eta}}(\boldsymbol{z}|\boldsymbol{x})}[f_{\boldsymbol{\phi}}(g_{\boldsymbol{\psi}}(\boldsymbol{z}))]. \quad (7)
$$

We train both the VAE and the discriminator with Equations (5) and Equation (6) in an alternating way.

### 3.2.2. From GAN- to SAN-based discriminator

We apply the slicing adversarial network (SAN) framework [31] to further enhance the motion VAE, based on a prior report showing SAN-based models perform better than the GAN counterparts. The impact of this replacement on motion generation is discussed in Section 4.

### 3.2.3. Architectures

We use a standard CNN-based architecture in the motion VAE encoder $q_{\boldsymbol{\eta}}$, decoder $g_{\boldsymbol{\psi}}$ and discriminator $f_{\boldsymbol{\psi}}$, consisting of 1D convolution, a residual block, and Leaky ReLU. For temporal downsampling and upsampling, we use stride 2 convolution and nearest interpolation, respectively. Specifically, the motion sequence $\boldsymbol{x} \in \mathbb{R}^{N \times L}$ is encoded into a latent vector $\boldsymbol{z} \in \mathbb{R}^{d_z \times d_l}$ with downsampling ratio of $d_l = L/4$. This architecture is inspired by [6, 38].

## 3.3. Stage 2: Motion Latent Diffusion

### 3.3.1. Text-conditional motion generation

In this section, we train a text-conditioned diffusion model on the low-dimensional motion latent space obtained by the autoencoder learned in the stage 1 (Section 3.2). Using the trained model, we perform motion generation conditioned on text. First, we define a time-dependent sequence $\boldsymbol{z}_0, \boldsymbol{z}_1, \ldots, \boldsymbol{z}_t, \ldots, \boldsymbol{z}_T \in \mathbb{R}^{d_z \times d_l}$ (starting from the VAE encoder output $\boldsymbol{z}_0 = \boldsymbol{z} \sim q_{\boldsymbol{\eta}}(\boldsymbol{z}|\boldsymbol{x})$), which is derived from the following Markov diffusion process in the

latent space: $q(\boldsymbol{z}_t|\boldsymbol{z}_{t-1}) = \mathcal{N}(\sqrt{\alpha_t}\boldsymbol{z}_{t-1}, (1-\alpha_t)\boldsymbol{I})$, where $T > 0$ and the constant $\alpha_t \in (0, 1)$ is a pre-defined noise-scheduling parameter that determines the forward process. The forward process allows for the sampling of $\boldsymbol{z}_t$ at an arbitrary time step $t$ in a closed form: $\boldsymbol{z}_t = \sqrt{\bar{\alpha}_t}\boldsymbol{z}_0 + \sqrt{1 - \bar{\alpha}_t}\epsilon$, where $\bar{\alpha}_t := \prod_{s=1}^{t}\alpha_s$ and $\epsilon \sim \mathcal{N}(\boldsymbol{0}, \boldsymbol{I})$.

As our goal is text-to-motion generation, our interest is in the conditional distribution $p(\boldsymbol{z}|\boldsymbol{c})$ given the text prompt $\boldsymbol{c}$. Here, similar to many text-conditioned latent diffusion models [2, 27], we train the conditional model $\epsilon_\theta(\boldsymbol{z}_t, t, \tau(\boldsymbol{c}))$ conditioned on the output of a text encoder $\tau(\boldsymbol{c})$, using the following objective function:

$$\mathcal{J}_{\text{cLDM}}(\theta) = \mathbb{E}_{\{\boldsymbol{z}_0, \boldsymbol{c}\}, \epsilon, t}\left[\|\epsilon - \epsilon_\theta(\boldsymbol{z}_t, t, \tau(\boldsymbol{c}))\|_2^2\right], \quad (8)$$

where $\boldsymbol{z}_0$ and $\boldsymbol{c}$ are drawn from the joint empirical distribution. In addition, as done in prior works, we adopt classifier-free guidance [13] and train the model unconditionally, i.e., without a text prompt, with a certain probability during training.

During inference, the trained diffusion model $\epsilon_\theta(\boldsymbol{z}_t, t, \tau(\boldsymbol{c}))$ is used to generate $\boldsymbol{z}_0$ through a denoising process conditioned on the text prompt $\boldsymbol{c}$. We adopt the sampling scheme of DDIM [28] with trailing sample steps [20], in which each sampling step is defined as:

$$\boldsymbol{z}_{t-1} = \sqrt{\bar{\alpha}_{t-1}}\left(\frac{\boldsymbol{z}_t - \sqrt{1 - \bar{\alpha}_{t-1}}\epsilon_\theta(\boldsymbol{z}_t, t, \tau(\boldsymbol{c}))}{\sqrt{\bar{\alpha}_t}}\right)$$
$$+ \sqrt{1 - \bar{\alpha}_{t-1} - \sigma_t^2}\epsilon_\theta(\boldsymbol{z}_t, t, \tau(\boldsymbol{c})) + \sigma_t\epsilon, \quad (9)$$

where $\sigma_t > 0$ determines the stochasticity of the sampling process, and the sampling process becomes deterministic when $\sigma_t = 0$. The part $(\boldsymbol{z}_t - \sqrt{1 - \bar{\alpha}_t}\epsilon_\theta(\boldsymbol{z}_t, t, \tau(\boldsymbol{c})))/\sqrt{\bar{\alpha}_t}$ in the first term corresponds to a direct estimate of the clean latent $\boldsymbol{z}_0$ from the noisy sample $\boldsymbol{z}_t$ using the diffusion model based on Tweedie's formula [5]; this estimate is denoted as $\boldsymbol{z}_{0|t}$. In actual inference, we use the estimated $z_0$ and the VAE decoder trained in stage 1 to obtain $g_\psi(\boldsymbol{z}_0)$ as the generated motion sequences. Variable length motion generation is then achieved by clipping a part of $g_\psi(\boldsymbol{z}_0)$ where the activation variable $a^i$ introduced in Section 3.1 satisfies $a^i < \delta$ ($0 < \delta < 1$). The effect of this approach is discussed in Section 4.1.

### 3.3.2. Architecture

We employ a diffusion transformer (DiT)-based architecture in our stage 2 model. The transformer used follows a standard structure of stacked blocks consisting of an attention layer and gated multilayer perceptrons (MLP) connected in series, with skip connections around each. Additionally, layer normalization is employed on the inputs of both the attention layer and the MLP. At the input and output of the transformer, linear mapping is used to convert from the latent dimension of the stage 1 model to the embedded dimension of the transformer. Text CLIP embedding is also added as input to the transformer, along with an embedding describing the current time step of the diffusion process. This architecture is inspired by [7], which established SOTA performance in audio generation.

### 3.4. Controllable Motion Generation on Latent Diffusion Sampling

In this section, we present a guided generation framework that leverages the pre-trained motion latent diffusion model (Section 3.3) for conditional motion generation and editing tasks without extra training. Major training-free methods (e.g, [3, 12, 37]) are based on the fact that the conditional score function can be decomposed into two additive terms: the unconditional score function and the log-likelihood term. Specifically, for a new condition $\boldsymbol{y}$, we have $\nabla_{\boldsymbol{z}_t}\log p(\boldsymbol{z}_t|\boldsymbol{c}, \boldsymbol{y}) = \nabla_{\boldsymbol{z}_t}\log p(\boldsymbol{z}_t|\boldsymbol{c}) + \nabla_{\boldsymbol{z}_t}\log p(\boldsymbol{y}|\boldsymbol{z}_t, \boldsymbol{c})$, which is derived from Bayes' rule. The conditional generation based on the aforementioned property can be regarded as a sequential procedure implemented as follows. First, a denoised sample $\boldsymbol{z}_{t-1}$ is obtained from $\boldsymbol{z}_t$ by the sampling step in Equation (9), without considering the given condition $\boldsymbol{y}$ (the first term in Equation (9)). Subsequently, $\boldsymbol{z}_{t-1}$ is further updated using the gradient of the log-likelihood term with respect to $\boldsymbol{z}_t$ (the second term).

The challenge here is that the log-likelihood term is computed based on noisy samples $\boldsymbol{z}_t$. In the classifier-guided diffusion [30], time-dependent classifiers for $\boldsymbol{z}_t$ have to be trained, which requires additional training. In contrast, in training-free methods, this term is approximated with the current clean data estimate $\boldsymbol{z}_{0|t}$ and a loss function $\mathfrak{L}(\boldsymbol{x}; \boldsymbol{y})$ defined for clean data. This loss function can be flexibly set depending on the task. For example, in inverse problems of the form $\boldsymbol{y} = \mathcal{A}(\boldsymbol{x})$, where $\mathcal{A}$ is a differentiable function with respect to $\boldsymbol{x}$, the loss function can be set as $\|\boldsymbol{y} - \mathcal{A}(\boldsymbol{x}_{0|t})\|_2^2$, where $\boldsymbol{x}_{0|t} = g_\psi(\boldsymbol{z}_{0|t})$ is a clean data estimate in the original data domain and obtained through the VAE decoder. Here, we adopt MPGD [12], a fast yet high-quality guidance method applicable to latent diffusion models. Following the denoising step, it updates the denoised sample based on the loss function $\mathfrak{L}(\boldsymbol{x}; \boldsymbol{y})$ as follows:

$$\boldsymbol{z}_{t-1} \leftarrow \boldsymbol{z}_{t-1} - \rho_t\sqrt{\bar{\alpha}_{t-1}}\nabla_{\boldsymbol{z}_{0|t}}\mathfrak{L}(g_\psi(\boldsymbol{z}_{0|t}); \boldsymbol{y}), \quad (10)$$

where $\rho_t$ is a time-dependent step size parameter.

More specifically, in editing motion tasks we generate motions to match given specific poses or trajectory control signals. To deal with various motion editing tasks in this framework, the loss function is designed as follows:

$$\mathfrak{L}_{\text{Motion}}(g_\psi(\boldsymbol{z}_{0|t}); \boldsymbol{y}) = \Sigma_n\Sigma_l m_{nl}\|\mathfrak{R}(g_\psi(\boldsymbol{z}_{0|t}))_{nl} - y_{nl}\|_2, \quad (11)$$

where $n$ and $l$ are indices of joint and frame, respectively, and $m_{nl}$ is a binary value indicating whether the control position $y_{nl}$ contains a valid value at frame $l$ for joint $n$, and $\Re(\cdot)$ is a function that converts the motion features including the joint's local positions to global absolute locations. We set $\mathcal{L}_{\text{Motion}}$ for loss function $\mathfrak{L}$ in the guided generation to measure the distance between desired constraints $\boldsymbol{y}$ and the joint locations of the generated motion. Target locations as constraint $\boldsymbol{y}$ can be specified for any subset of joints in any subset of motion frames.[2] Editing a generated motion to match specific poses or follow a specific trajectory is achieved by minimizing $\mathcal{L}_{\text{Motion}}$ using the update rule in Equation (10).

## 4. Experiments

We evaluate the performance of MoLA on two tasks: motion generation (Section 4.1) and motion editing (Section 4.2). Our results demonstrate that MoLA achieves its three key objectives: (1) fast and high-quality generation, (2) variable-length generation, and (3) multiple motion editing tasks in a training-free manner. To validate these objectives, we utilize the HumanML3D [10].[3]

### 4.1. Motion Generation

#### 4.1.1. Performance comparison with other text-to-motion models

We evaluate our proposed MoLA in comparison to current SOTA methods [2, 4, 11, 19, 22, 24, 25, 32, 33, 38, 40, 42, 43] using five metrics (R-Precision, Fréchet Inception Distance (FID), Multi-modal distance (MMDist), Diversity, and MultiModality (MModality)) proposed by Guo *et al.* [10]. For evaluation, we select the model that achieves the best FID, which is a metric that evaluates the overall motion quality, on the validation set and report its performance on the test set of HumanML3D. We show the results in Table 1. The methods are organized into three groups: i) those using VQ-based latent representations (**Discrete**), ii) those using data-space diffusion model (**Continuous (raw data)**), and iii) those using VAE-based latent representations (**Continuous (latent)**). The discrete approaches perform well in motion generation (e.g., [11, 22, 25]). However, those models cannot control an arbitrary set of joints in a training-free manner [25] as we have discussed so far. MDM [32], MotionDiffuse [40] and Fg-T2M [33] adopt

---

[2]As examples of motion editing, if $\boldsymbol{y}$ is given as the start-end positions, we can handle the motion in-betweening. If $\boldsymbol{y}$ is given as the lower body positions, we can edit the upper body corresponding to the lower one. If $\boldsymbol{y}$ is given as the pelvis trajectory, it corresponds to the path-following task. We leave the task details to Section 4.2. Note that the guided generation framework in Equations (10) and Equation (11) has the potential to generate motion while dealing with a variety of time and spatial constraints not limited to these three task examples.

[3]This dataset contains 14,616 human motions from the AMASS [23] and HumanAct12 [9] datasets and 44,970 text descriptions.

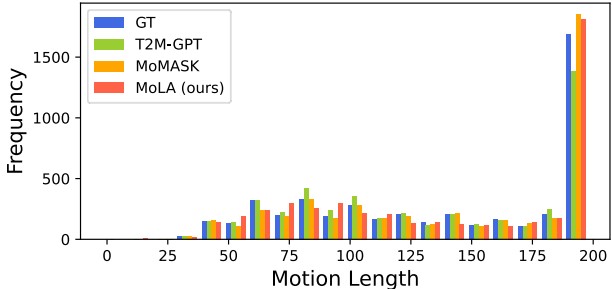

Figure 4. Comparison of motion length distributions between the HumanML3D test set and the generated motion samples. The Jensen-Shannon divergence (JSD) for each distribution is as follows: JSD(GT||T2M-GPT) = 0.041, JSD(GT||MoMASK) = 0.040, and JSD(GT||MoLA) = 0.026. Similarly, the Earth Mover's Distance (EMD) for each distribution is given by $\mathcal{D}_{\text{EMD}}(\text{GT}, \text{T2M-GPT}) = 6.706$, $\mathcal{D}_{\text{EMD}}(\text{GT}, \text{MoMASK}) = 3.673$, and $\mathcal{D}_{\text{EMD}}(\text{GT}, \text{MoLA}) = 3.538$ (a unit in this EMD means 1 frame).

data-space diffusion, while MLD [2], MotionLCM [4] and MoLA utilize a diffusion model in a lower-dimensional latent space. Therefore, the former are grouped in Continuous (raw data) and the latter in Continuous (latent) in the table. MoLA achieves the best performance in continuous methods, especially in the R-precision, FID, and MMDist metrics as shown in Table 1. Moreover, we also evaluate generation quality and inference cost in comparison to existing text-to-motion methods that have publicly available implementations [2, 11, 25, 32, 38]. As shown in Figure 2, MoLA is much faster in generation than MDM, which is the only existing method that performs training-free motion editing.

#### 4.1.2. Variable-length motion generation

Next, we discuss the impact of the activation variable in the motion representation introduced in Section 3.1. As shown in Equation (2), incorporating the activation variable into the motion features enables the Stage 2 model (Section 3.3) to generate variable-length motions. Figure 4 shows a comparison of the length distributions of samples generated by two existing methods (T2M-GPT and MoMask) and MoLA against the test set of HumanML3D. T2M-GPT represents a typical auto-regressive Text-to-Motion approach, while MoMask is a mask-prediction-based method that requires specifying the motion length. To address this limitation, MoMask introduces a length estimator that generates motion lengths conditioned on the input text. From Figure 4, we observe that our method successfully generates variable-length motions. Furthermore, the distribution of motion lengths generated by MoLA is closer to the actual motion distribution than those of the two existing methods, demonstrating the effectiveness of our approach.

| Category | Method | R-Precision ↑ | | | FID ↓ | MMDist ↓ | Diversity → | MModality ↑ |
|---|---|---|---|---|---|---|---|---|
| | | Top-1 | Top-2 | Top-3 | | | | |
| N/A | Real motion data | $0.511^{\pm.003}$ | $0.703^{\pm.003}$ | $0.797^{\pm.002}$ | $0.002^{\pm.000}$ | $2.974^{\pm.008}$ | $9.503^{\pm.065}$ | - |
| **Discrete** | M2DM [19] | $0.497^{\pm.003}$ | $0.682^{\pm.002}$ | $0.763^{\pm.003}$ | $0.352^{\pm.005}$ | $3.134^{\pm.010}$ | $9.926^{\pm.073}$ | $3.587^{\pm.072}$ |
| | AttT2M [42] | $0.499^{\pm.003}$ | $0.690^{\pm.002}$ | $0.786^{\pm.002}$ | $0.112^{\pm.006}$ | $3.038^{\pm.007}$ | $9.700^{\pm.090}$ | $2.452^{\pm.051}$ |
| | T2M-GPT [38] | $0.492^{\pm.003}$ | $0.679^{\pm.002}$ | $0.775^{\pm.002}$ | $0.141^{\pm.005}$ | $3.121^{\pm.009}$ | $9.722^{\pm.081}$ | $1.831^{\pm.048}$ |
| | MoMask [11] | $0.521^{\pm.002}$ | $0.713^{\pm.002}$ | $0.807^{\pm.003}$ | $0.045^{\pm.002}$ | $2.958^{\pm.008}$ | - | $1.241^{\pm.040}$ |
| | DiverseMotion [22] | $0.496^{\pm.004}$ | $0.687^{\pm.004}$ | $0.783^{\pm.003}$ | $0.070^{\pm.004}$ | $3.063^{\pm.011}$ | $9.551^{\pm.068}$ | $2.062^{\pm.079}$ |
| | MMM [25] | $0.504^{\pm.003}$ | $0.696^{\pm.003}$ | $0.794^{\pm.002}$ | $0.080^{\pm.003}$ | $2.998^{\pm.007}$ | $9.411^{\pm.058}$ | $1.164^{\pm.041}$ |
| | ParCo [43] | $0.515^{\pm.003}$ | $0.706^{\pm.003}$ | $0.801^{\pm.002}$ | $0.109^{\pm.003}$ | $2.927^{\pm.008}$ | $9.576^{\pm.088}$ | $1.382^{\pm.060}$ |
| | BAMM [24] | $0.525^{\pm.002}$ | $0.720^{\pm.003}$ | $0.814^{\pm.003}$ | $0.055^{\pm.002}$ | $2.919^{\pm.008}$ | $9.717^{\pm.089}$ | $1.687^{\pm.051}$ |
| **Continuous (data space)** | MotionDiffuse [40] | $0.491^{\pm.001}$ | $0.681^{\pm.001}$ | $0.782^{\pm.001}$ | $0.630^{\pm.001}$ | $3.113^{\pm.001}$ | $9.410^{\pm.049}$ | $1.553^{\pm.042}$ |
| | MDM [32] | $0.320^{\pm.005}$ | $0.498^{\pm.004}$ | $0.611^{\pm.007}$ | $0.544^{\pm.044}$ | $5.566^{\pm.027}$ | $9.559^{\pm.086}$ | $2.799^{\pm.072}$ |
| | Fg-T2M [33] | $0.492^{\pm.002}$ | $0.683^{\pm.003}$ | $0.783^{\pm.002}$ | $0.243^{\pm.019}$ | $3.109^{\pm.007}$ | $9.278^{\pm.072}$ | $1.614^{\pm.049}$ |
| **Continuous (latent)** | MLD [2] | $0.481^{\pm.003}$ | $0.673^{\pm.003}$ | $0.772^{\pm.002}$ | $0.473^{\pm.013}$ | $3.196^{\pm.010}$ | $9.724^{\pm.082}$ | $\mathbf{2.413^{\pm.079}}$ |
| | MotionLCM [4] | $0.502^{\pm.003}$ | $0.698^{\pm.002}$ | $0.798^{\pm.002}$ | $0.304^{\pm.012}$ | $3.012^{\pm.007}$ | $9.607^{\pm.066}$ | $2.259^{\pm.092}$ |
| | MoLA (ours) | $\mathbf{0.516^{\pm.006}}$ | $\mathbf{0.712^{\pm.005}}$ | $\mathbf{0.805^{\pm.004}}$ | $\mathbf{0.115^{\pm.004}}$ | $\mathbf{3.008^{\pm.016}}$ | $\mathbf{9.885^{\pm.152}}$ | $2.156^{\pm.157}$ |

Table 1. Comparison with state-of-the-art methods on HumanML3D dataset. Note that discrete representations do not allow for training-free motion editing; therefore, methods based on VQ-based latent representations (Discrete) are grayed out. The best scores for each metric in the methods using VAE-based latent representations (Continuous (latent)) are highlighted in **bold**.

| Editing type | Methods | R-Precision Top-3 ↑ | FID ↓ | Diversity → | Traj. err. ↓ | Loc. err. ↓ | Avg. err. ↓ | AITS ↓ |
|---|---|---|---|---|---|---|---|---|
| **Training-based editing** | OmniControl | 0.688 | 0.192 | 9.533 | 0.065 | 0.007 | 0.053 | 74.4 |
| | MotionLCM | 0.759 | 0.501 | 9.293 | 0.237 | 0.054 | 0.164 | 0.02 |
| **Training-free editing** | MoLA (ours) | 0.761 | 0.486 | 9.322 | 0.271 | 0.051 | 0.159 | 1.04 |

Table 2. Comparison of motion editing (path-following task) on HumanML3D dataset

## 4.2. Motion Editing

Here, we demonstrate three types of editing tasks using a unified framework: path-following (motion guided by a specified trajectory), in-betweening (editing in the time direction), and upper-body editing (modifying specific joints). In particular, for path-following, we quantitatively compare our approach with existing models [4, 35].

**Path-following** is a task of giving a trajectory (often the position of the pelvis) and generating the motion that matches the given route. Controlling the trajectory of generated motion enables more motion variation, avoidance of obstacles, and the creation of motion that meets physical constraints. In this experimental case, the desired pelvis trajectory is set as the control signal $y$ in Equation (11). The upper row of Figure 5 shows the editing results with our model when different path controls are given as $y$ in the same text condition. In addition, we show a quantitative comparison with existing methods: OmniControl [35] and MotionLCM [4], where the former is Continuous (raw data) type method and the latter is Continuous (raw latent) one. We compare them in terms of FID, R-Precision, Diversity, Trajectory err (50cm), Location err (50cm), Avgerage err, and Average inference time per sentence (AIST) in Table 2, following [4]. Table 2 demonstrates that MoLA achieves competitive editing performance across all compared to other editing methods. Notably, while MoLA shows lower performance in following control signals compared to OminiControl, it achieves significantly faster edit-

ing. Note that OmniControl and MotionLCM have been trained for this specific editing task, while MoLA performs path-following without model fine-tuning. In other words, MoLA can perform different tasks without additional training as shown in the following sections, but OmniControl and MotionLCM require training a separate model for each task. MoLA is a more flexible and efficient framework.

**In-betweening** is an important editing task that interpolates or fills the gaps between keyframes or major motion joints to create smooth 3D motion animation. Our framework can coordinate and generate motion between past and future contexts without additional training. We only need to set the start-end positions or the motions of a few frames as the control signal $y$ in Equation (11). The middle row of Figure 5 depicts the motion in-betweening results with our model when different start-end controls are given as $y$ in the same text condition.

**Upper body editing** combines generated upper body parts with given lower body parts. Generating some joints while keeping the other body joints following a given control signal can be seen as the task of outpainting in the spatial dimension of motion. The control signal $y$ in Equation (11) is set to lower body positions that are not subject to editing.[4] The lower row of Figure 5 shows the upper body editing results with our model when different lower body controls are given as $y$ in the same text condition.

---

[4]Note that, although we are dealing with upper body editing in this experiment, it is in principle possible to specify a different joint subset.

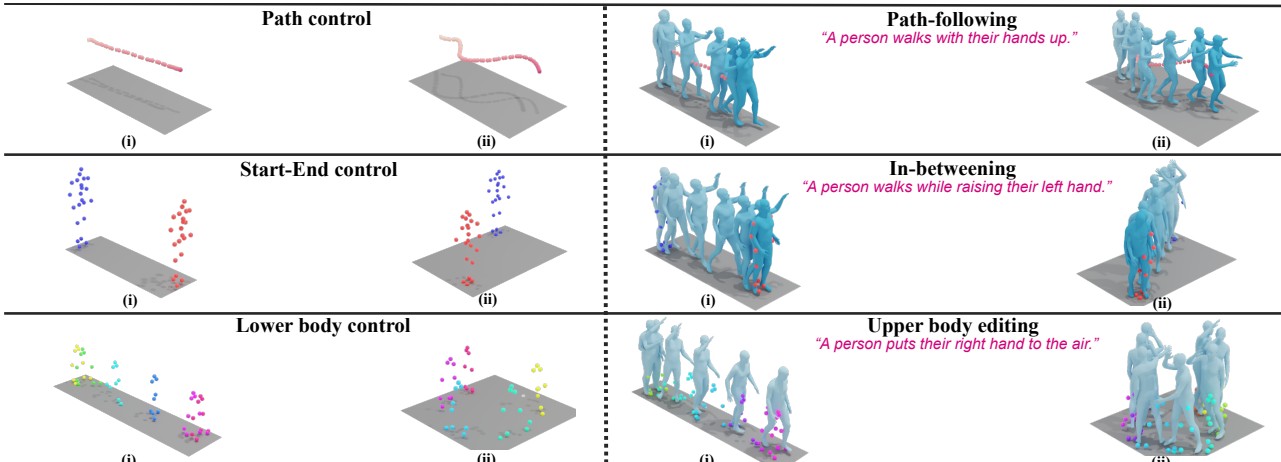

Figure 5. Qualitative results for the three editing tasks. For the three motion editing tasks (path following, upper-body editing, and in-betweening), we treat each control signal (i) and (ii) in the left side of the figure as $\boldsymbol{y}$ in Equation (10) and (11). The corresponding generated results using the same input text are shown on the right side of the figure as (i) and (ii), respectively.

| Method | Reconstruction | | Generation | |
|---|---|---|---|---|
| | rFID ↓ | MPJPE ↓ | FID ↓ | MMDist ↓ |
| Dimension of latent space | | | | |
| MoLA ($d_z = 8$) | 0.110 | 54.2 | 0.183 | 3.099 |
| **MoLA** ($d_z = 16$) | 0.030 | 29.3 | **0.115** | **3.008** |
| MoLA ($d_z = 32$) | **0.028** | **26.8** | 0.904 | 3.536 |
| Adversarial training | | | | |
| *w/o* GAN or SAN | 0.038 | 29.3 | 0.126 | 3.053 |
| *w/* GAN instead of SAN | 0.032 | 29.5 | 0.141 | 3.044 |
| Input for the encoder $q_\eta$ | | | | |
| $[(\boldsymbol{m}^i)^\mathsf{T}, a^i]^\mathsf{T}$ instead of Eq. (2) | 0.029 | 31.2 | **0.112** | 3.024 |

Table 3. Analysis of motion reconstruction and generation performance on HumanML3D dataset

| Method | Traj. err. ↓ | Loc. err. ↓ | Avg. err. ↓ |
|---|---|---|---|
| Input for the encoder $q_\eta$ | | | |
| $[(\boldsymbol{m}^i)^\mathsf{T}, a^i]^\mathsf{T}$ instead of Eq. (2) | 0.281 | 0.068 | 0.174 |
| **MoLA** ($d_z = 16$) | **0.271** | **0.051** | **0.159** |

Table 4. Analysis of motion editing (path-following task) performance on HumanML3D dataset

## 4.3. Ablation studies

We discuss the impact of latent space dimensionality, adversarial training, and motion representation on the stage 1 model. The dimensionality of the latent space may affect not only reconstruction quality but also influence the difficulty of training in stage 2, ultimately impacting the quality of the generated outputs. To investigate this, we evaluate the performance for cases with $d_z = \{8, 16, 32\}$. Table 3 shows that although the case of $d_z = 16$ does not achieve the best reconstruction quality compared to the other cases, it performs best in terms of FID and MMDist. Therefore, we adopted $d_z = 16$ for MoLA. The results for VAE combined with GAN/SAN and modifying the encoder inputs are also shown in Table 3. As shown in Table 3, the adversarial training is effective in the motion reconstruction task. It not only improves the reconstruction performance in stage 1 but also contributes to enhanced generation performance in stage 2. In particular, we can improve the performance of the stage 1 model by adopting the SAN framework instead of the conventional GAN. A better rFID is directly related to the upper bound of the overall performance of a text-to-motion model.

Furthermore, as shown in Tables 3 and 4, modifying the encoder input Equation (2) improves reconstruction quality in terms of MPJPE without significantly compromising generation quality. As a result, this modification has a beneficial effect on the performance of the motion editing task, which requires fitting motion sequences into given control signals. This improvement can be attributed to the quality of the decoder $g_\psi$, which contributes to the performance of the editing task, as indicated by Equations (10) and (11). Thus, we adopted VAE trained with the SAN framework [31] and the motion representation Equation (2) for MoLA.

## 5. Conclusion

We proposed MoLA, a text-to-motion model that achieves fast, high-quality generation with multiple control tasks in a single framework. We rethought the motion representation and introduced an activation variable that characterizes the length of the motion. In addition, we integrated latent diffusion, adversarial training, and a guided generation framework. Our experiments demonstrated MoLA's ability to generate variable-length motions with distributions close to real motions and perform diverse motion editing tasks, significantly extending the performance boundaries of methods categorized as enabling training-free editing.

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
