# OpenReview forum: "MoLA: Motion Generation and Editing with Latent Diffusion Enhanced by Adversarial Training"
_thecvf.com/CVPR/2025/Workshop/HuMoGen — CVPR 2025 Workshop HuMoGen Submission_

### Official Review · Reviewer_Z4cF · 2025-03-26
**Adapting Image Diffusion Techniques for Motion: Solid Execution, Limited Novelty**

**Rating:** 4
**Confidence:** 4

**Review:**

# Summary
This paper presents a new text-to-human motion generation model based on a latent diffusion architecture.
It introduces adversarial training during the VAE stage - an approach well-known in other domains, such as image generation, but not previously applied to motion generation. Furthermore, the model includes an activation variable (indicator/mask for frames) for the motion representation that allows generation of variable-length motion, and incorporates MPGD (Manifold Preserving Guided Diffusion) guidance for training-free motion editing. While the proposed model does not outperform state-of-the-art VQ-VAE based auto-regressive models in general generation quality, it surpasses models capable of training-free motion editing in performance.

# Strengths
 - Novel application of adversarial VAE training in the motion generation domain, adapted from image synthesis literature.
 - Incorporation of an indicator variable for flexible, non-fixed-length motion generation is a practical design choice.
 - Supports training-free motion editing via MPGD guidance, enabling powerful motion editing capabilities.
 - The paper is written clearly, and is easy to follow.
 - Competitive performance compared to motion-generation models that use continuous representation.
 - SOTA performance for training-free motion editing.

# Weaknesses
- Contributions are adaptations from other domains, primarily image generation, rather than new algorithmic ideas developed for motion.
- Even though fast inference times is listed as an advantage several times, the paper lacks a through evaluation of inference time for general motion generation against SOTA models, and only compares inference time against 2 training-based motion editing methods.
- The models doesn't outperform SOTA human motion generation auto-regressive methods using VQ-VAE based discreet representation.

# Evaluation

## Quality
The paper is technically sound and evaluates its model across standard benchmarks. Performance on general motion generation lags a bit behind state-of-the-art, but performance of training-free motions editing is very good.

## Clarity
The paper is well-written, well-structured, and effectively communicates its ideas and methods.

## Originality
While the adaptation of adversarial VAE training and MPGD guidance to motion generation is interesting, these techniques are not new and their novelty lies mainly in domain transfer.

## Significance
The paper shows how known techniques can be applied successfully to motion generation, particularly for training-free editing, which is a valuable contribution. However, the limited performance gains and lack of novel techniques limits its impact.

---

### Official Review · Reviewer_FRiX · 2025-03-26
**Review of MoLA**

**Rating:** 4
**Confidence:** 4

**Review:**

## Summary:
The paper presents MoLA which is a text-to-motion generation framework based on latent-diffusion models. This framework allows for fast and variable length motion generation and provides training-free editing control over the latent-based generation.

## Strengths:
- Paper is easy to read and presentation is clear.
- The improvement in inference time is quite large (Fig 2)
- The experiment section is thorough and it addresses most of the claimed contributions from introduction/method.

## Weaknesses:
- Results in video (@00:46-01:00) show jittery motion. This might be due to the removal of velocity in input representation or something else in the architecture (not discussed).

- One of the contributions of the paper is variable length generation. Authors evaluate this by comparing motion length distribution between GT test set and predictions from MoLA. However, since textual inputs also affect each variable length generation differently, the authors should qualitatively evaluate or discuss how generations of different lengths are affected by varying lengths of text. Does having variable-length output improve motion appropriateness for long or compositional text prompts?


## Minor Weaknesses:
- Regarding Ablation of Eq.(2) $[(\mathbf{{m}}^i)^\top, a^i]^\top$: Authors should have calculated foot contact loss and velocity difference (or smoothness error ) instead of calculating MPJPE for all the joints. This would actually evaluate whether removing the information (as done in Eq.(2)) adversely affects the reconstruction quality.

- In Table 2. it can be seen that Traj. err, Loc. err are higher for methods which perform editing in latent space. I believe this subtle point should have been clearly discussed during analysis.

### Typos or Writing Improvement suggestions:
- Ln. 158 "high generation performance" -> "high *quality* generation performance"?
- Ln. 199 Missing citation to VAE,  Ln 329 Missing citation to CLIP
- Ln. 106-109: maybe improve sentence structure at this line
- Supplementary document Figure 6 x-axis label: scase -> scale

---

### Meta-Review · Area_Chair_KpW7 · 2025-03-30

**Recommendation:** Accept

**Metareview:**

This paper presents MoLA, a text-to-motion generation model leveraging latent diffusion, adversarial VAE training, and MPGD guidance for training-free motion editing. The framework achieves fast inference and supports variable-length motion generation with an activation variable. The paper is well-written and provides a thorough experimental evaluation, demonstrating strong results in training-free motion editing. While inference speed is highlighted as a key contribution, it may not achieve the same quality as the sota models. Additionally, minor concerns include jittery motion artifacts and the need for a deeper analysis of variable-length motion generation. Despite these limitations, the paper offers a valuable contribution to motion generation, particularly for training-free editing. As a result, I recommend acceptance for this paper.

---

### Decision · Program_Chairs · 2025-03-31

Accept